# Whole-Genome Sequencing Reveals the Genomic Characteristics and Selection Signatures of Hainan Black Goat

**DOI:** 10.3390/genes13091539

**Published:** 2022-08-26

**Authors:** Qiaoling Chen, Yuan Chai, Wencan Zhang, Yiwen Cheng, Zhenxing Zhang, Qi An, Si Chen, Churiga Man, Li Du, Wenguang Zhang, Fengyang Wang

**Affiliations:** 1Hainan Key Lab of Tropical Animal Reproduction, Breeding and Epidemic Disease Research, Animal Genetic Engineering Key Lab of Haikou, College of Animal Science and Technology, Hainan University, Haikou 570228, China; 2College of Agronomy, Animal Husbandry and Bioengineering, Xing’an Vocational and Technical College, Ulanhot 137400, China; 3College of Animal Science, Inner Mongolia Agricultural University, Hohhot 010010, China

**Keywords:** Hainan Black goat, whole-genome resequencing, genetic diversity, selective signal

## Abstract

Goats have become one of the most adaptive and important livestock species distributed in developing countries in recent years. The Hainan Black goat is a native goat breed of the Hainan region that is generally well-liked by the local population and is thus raised in large numbers. However, the genomic diversity and selective signals of the Hainan Black goat have not been clearly elucidated yet. Therefore, in this study, we performed whole-genome resequencing of 16 Hainan Black goats and compared the results with those of 71 goats of 6 other breeds from different geographic regions. Principal component analysis (PCA) and phylogenetic analysis identified seven lineages for all goats. Hainan Black goats showed the most similarity with Leizhou goats and the least similarity with Boer goats. Selective sweep analysis identified candidate genes associated with various functions, including immune resistance to disease (*TNFAIP2* (TNF alpha induced protein 2) and *EXOC3L4* (exocyst complex component 3 like 4)), melanin biosynthetic process (*CDH15* (cadherin 15)*, ASIP* (agouti signaling protein)*,* and *PARD3* (par-3 family cell polarity regulator)), and light sensitivity (*CNGB3* (cyclic nucleotide gated channel subunit beta 3) and *CNBD1* (cyclic nucleotide binding domain containing 1)), underlying strong selection signatures in Hainan Black goats. The melanin biosynthetic process, circadian entrainment, regulation of cyclic adenosine 3,5-monophosphate (cAMP)-mediated signaling, and the Rap-1 signaling pathway were significantly enriched in Hainan Black and Alashan Cashmere goats. This result may be important for understanding each trait. Selection signature analysis revealed candidate single nucleotide polymorphisms (SNPs) and genes correlated with the traits of Hainan Black goats. Collectively, our results provide valuable insights into the genetic basis of specific traits correlated with the Hainan island climate, artificial selection in certain local goat breeds, and the importance of protecting breed resources.

## 1. Introduction

Asia has the largest world population of goats [1]. In particular, China has abundant goat resources and a large goat population due to the presence of diverse ecosystems. In 2020, the total number of goats being raised in China was 133.45 million [2]. The distribution regions of goats, according to the ecogeographical conditions in China, are as follows: the north-east agricultural, Inner Mongolia, North China, north-west agro-pastoral, Xinjiang pastoral, south-central agricultural, south-west agricultural, and Qinghai–Tibet plateau regions [3]. However, most goat populations are distributed in the southern grassy mountain slopes and the arid, barren desert areas. Hainan is located in the south-central agricultural region, which has a natural environment with a tropical island climate. The Hainan Black goat, related to the Leizhou goat, is an excellent local goat breed bred for meat; it has the following characteristics: (1) better tolerance in a hot and humid climate [4]; (2) tolerance for crude feed; (3) tender and delicious meat without uncomfortable flavor [5]. However, studies on the genetic characteristics of Hainan Black goats are limited.

With the rapid development of modern sequencing technology, whole-genome sequencing (WGS) has enabled researchers to simultaneously examine sequence differences among populations, morphs, and breeds for hundreds of genes and gene families, especially well-studied species with large genome sizes [6]. Li et al. identified 135 genomic regions associated with Cashmere fiber traits within Cashmere goat populations [7]. Several genes that may be associated with goat hair color, altitude adaptation, growth, and reproductive characteristics have also been identified [8]. By analyzing the genomes of goats worldwide, Zheng et al. provided evidence of an ancient introgression event from a West Caucasian tur-like species to the ancestor of domestic goats [9]. Chen et al. used WGS to identify selective signals implicated in immune resistance to skin disease in Longlin goats [10].

Based on the literature, studies on the assessment of Hainan Black goats using WGS are limited. In this study, the entire genome of the Hainan Black goat was resequenced and analyzed. Further, five goat breeds, Alashan Cashmere, Dazu Black, Jining Grey, Longlin, and Leizhou, were selected from four typical regions of China, including Inner Mongolia, the south-west agricultural region, North China, and the south-central agricultural region. A foreign breed (Boer goat) from South Africa was selected as a control. Subsequently, the specific genetic characteristics of the different breeds were analyzed by comparing their whole-genome data. Our findings provide a theoretical basis for molecular-marker-assisted breeding selection technology and a reference for the excavation of genetic characteristics of the Hainan Black goat.

## 2. Materials and Methods

### 2.1. Animal Sampling and Whole-Genome Resequencing

Genomic DNA was collected from the ear tissues of 16 Hainan Black goats using the standard phenol–chloroform extraction method and subjected to electrophoresis (in 1.2% agarose gel) and UV spectrophotometry. A library was constructed using the TruSeq DNA PCR-free prep kit according to the manufacturer’s guidelines, and WGS was performed using the Illumina NovaSeq platform. Raw sequence data were deposited in GenBank (accession number PRJNA754269). Sequence reads were analyzed using the quality control tool, FastQC (https://www.bioinformatics.babraham.ac.uk/projects/fastqc/ (accessed on 4 January 2022), followed by quality filtering based on the sequence quality score and the presence of filtering reads shorter than 50 bp. Pair-end sequence reads were mapped to the reference goat genome (ARS1) [11] using BWA [12] with default parameters. Sequencing alignment (SAM) format files were imported to Picard (http://broadinstitute.github.io/picard/, version 1.92 (accessed on 4 January 2022)) for sorting and merging. We used Picard Tools to remove duplicate reads generated during the PCR amplification stage. Thereafter, a local realignment of the mapped reads around indels was performed using the GATK package [13]. The overall alignment rate of reads to the reference sequence was 99.48%.

A total of 71 publicly available genomes from 6 representative breeds were employed (raw data origin was in Appendix A). Resequencing data from 71 goat samples from the Inner Mongolia, south-west agricultural, North China, and south-central agricultural regions of China, as well as from the South Africa region, were downloaded from the National Center for Biotechnology Information (NCBI) website and divided into seven groups: Longlin goat (popA), Leizhou goat (popB), Hainan Black goat (popC), Dazu Black goat (popD), Jining Grey goat (popE), Boer goat (popF), and Alashan Cashmere goat (popG) (Table 1).

### 2.2. Single Nucleotide Polymorphism (SNP) Calling and Annotation

The initial round of variant calling using the original BAM files was performed using a unified genotyper in Genome Analysis Toolkit (GATK) [13] with stand-call-conf set to 20.0 and stand-emit-conf set to 10.0. SNPs were selected and filtered via hard filters on the raw variant calls (parameters for SNP filtering: FS ≤ 60, MQ ≥ 40, QD ≥ 2, HaplotypeScore ≤ 13, ReadPosRankSum ≥ −8, MQRankSum ≥ −12.5; parameters for indel filtering: FS ≤ 200, DP ≥ 4. QD ≥ 2, ReadPosRankSum ≥ −20). Gene-based analysis using ANNOVAR software [14] was used to functionally annotate the putative SNPs. For annotation, SNPs were classified on the basis of their protein-coding location (nonsynonymous or synonymous), their location (exonic, intronic, or intergenic; 5′UTR or 3′UTR; splice acceptor or donor site; downstream or upstream), and their functional annotation (stop codon gain or loss, and amino acid changes), which were determined by the goat reference genome (ARS1). Standard settings for the ANNOVAR analysis were used.

### 2.3. Phylogenetic and Population Structure Analyses

Clustering and principal components analysis (PCA) were performed using GCTA (http://www.complextraitgenomics.com/software/gcta/, accessed on 7 February 2022) to identify potential outliers in each dataset. Five subjects were detected as population outliers and were removed from the dataset. Samples were clustered separately based on the distribution status of the first principal component, excluding the outliers in some groups from further analysis. A phylogenetic tree was constructed using the maximum-likelihood algorithm of the fastTree software (http://www.microbesonline.org/fasttree/, accessed on 8 February 2022). After establishing the phylogenetic tree, the reliability of the branches (bootstrap, 1000 replications) was verified. A population structure analysis was performed based on admixture analysis (http://dalexander.github.io/admixture, accessed on 8 February 2022) to identify similarities and differences between the target population and other goat populations. The number of assumed ancestry population, K, ranged from one to ten. The remaining parameters were set as default values. Linkage disequilibrium (LD) decay was calculated using PopLDdecay v3.40 [15] for each breed and all individual goats.

### 2.4. Genome-Wide Analysis of Genetic Diversity and Detection of the Selective Sweeps

The average pairwise nucleotide diversity (θπ) and Tajima’s D statistics of each breed were calculated using a sliding window approach (20 kb sliding windows in 5 kb steps) with the default parameters of VCFtools [16]. Population differentiation was measured by pairwise Fst using the VCFtoools with the default parameters. To detect variations associated with selective sweeping, the θπ value and Fst values were plotted using GraphPad Prism 8 software (GraphPad, San Diego, CA, USA). The regions under selection in Hainan Black goats were annotated using a custom R script based on the goat reference genome (ASR1). Cluego software (http://apps.cytoscape.org/apps/Cluego, accessed on 15 February 2022), the Gene Ontology (GO) database (released March 2015), and the Kyoto Encyclopedia of Genes and Genomes (KEGG) database (released March 2015) were used to classify identified genes into specific functional terms and conduct pathway enrichment analysis. Gene ontology analysis based on biological processes and enrichment analysis was performed using the right-side hyper-geometric statistic test; the resulting probability value was then corrected using Bonferroni’s method. Protein–protein interaction (PPI) network analysis was performed to reveal gene interaction using the STRING database (https://string-db.org, accessed on 15 February 2022).

### 2.5. Population−Specific SNP Analysis and Gene Flow Analysis

Population−specific SNPs were identified based on the frequency of genotype distribution among populations. A significant difference was observed based on genotype frequency. Genotype frequency was calculated by dividing the number of individuals with a given genotype by the total number of samples: (i) homozygous ref genotype 0/0, (ii) homozygous ref type genotype frequency 0/1, and (iii) heterozygote alt genotype. If the genotype frequency was over the threshold of population−specific genotype frequency (0.8) in the “A” population and less than the threshold of population−specific genotype frequency (1–0.8) in the “B” population, the genotype was considered specific to population A. The TreeMix software [17] was used to model gene flow between seven populations using default settings. Bootstrapping with 1000 replications was performed for Treemix.

## 3. Results

### 3.1. Genetic Variation among Different Goat Breeds

In genetics, there has been considerable focus on characterizing rare variations in the goat populations and other important livestock species and the role played by these variants in genetic diversity and genomic footprints under positive selection to account for adaptations to the local environments [6,7,8,9,10,18,19]. Goat samples from seven breeds were selected in the present study (Figure 1). The main purpose of this study was to systematically identify the selection signals underlying phenotypic evolution in goats using SNPs obtained from whole-genome DNA resequencing data. A total of 1.6 billion reads with an average depth of 4.78× (4.25–5.23×) for 16 Hainan Black goat samples were obtained. Based on sequence detection, approximately 28,060,846–105,154,850 clean reads were obtained after quality control. After mapping the clean reads of all samples against the goat reference genome, an average mapping rate of 99.5% was found, indicating the high quality of the sequencing data (Table A1). After analyzing all sample data, a total of 88,454,696 SNPs and 87 samples remained in the final dataset for downstream analysis (Figure 2). The number of intronic and intergenic SNPs was markedly higher than that of exonic SNPs, up to 28,635,232 (32.37%) and 57,102,950 (64.56%), respectively. Further, 517,191 synonymous SNPs (0.58%) and 332,542 nonsynonymous SNPs (0.38%) were identified among the exonic SNPs. Compared to the reference genome, Hainan Black goats were detected to have the maximum number of homozygous and heterozygous genotypes consistent with the genome. Subsequently, all types of SNPs were analyzed. The SNP spectrum analysis results revealed T:A>C:G and C:G>T:A were the main mutation types (Figure 2B). In addition, a total of 4,498,168 indels were obtained from 87 goats in 7 breeds (Figure 2C). Most insertions and deletions were obtained in the Hainan Black goat.

### 3.2. Population Structure and Characterization of Hainan Black Goat

To examine the relationships among individuals, PCA was performed among all individuals. A total of five subjects were detected as population outliers and removed from the dataset. Linkage disequilibrium (LD) analysis revealed that the highest average LD (r^2^) was found in the Leizhou goat, while the lowest average LD was found in the Hainan Black goat (Figure 3A). The remaining 82 individuals in the dataset analysis represented two principal components (PC1 and PC2), with variances of 4.69% and 6.12%, respectively. PCA results showed that the Longlin goat had the most similarity with the Alashan Cashmere goat; the Dazu Black goat had the least similarity with the Jining Grey goat. Two individual Dazu Black goats were extremely different from other goats on the PCA plot (Figure 3C). A neighbor-joining tree was constructed based on the autosomal SNPs (Figure 3D). Leizhou goats were found to be genetically closer to Hainan Black goats. Further, the admixture analysis suggested K = 6 as the most likely number of genetically distinct groups within the 82 goat samples (Figure 3E), highlighting the different breeds of goats. According to population structure analysis, Hainan Black goats were closely related to Leizhou and Dazu Black goats. Maximum-likelihood approach methods were used to analyze migration events between the seven breeds (Figure 3B). The vector from popB to popC suggests gene flows from the Leizhou goat to the Hainan Black goat.

### 3.3. Genetic Diversity Analysis

To predict the genetic diversity of goats in each population, we evaluated the migration and reproduction of the different goat breeds. The expected heterozygosity (Exp Het) and observed heterozygosity (Obs Het) were calculated for each group. In each group, the Exp Het value was always observed to be higher than the Obs Het value, which indicates that inbreeding can occur in groups. In general, the Fis of a group has a positive value. As shown in the statistical results of the genetic diversity of each population (Table 2), each group exhibited stability of nucleotide diversity, which is reflected by similar low values of Pi. Compared with other breeds, the Hainan Black goats had the highest Num Indv, which is related to the number of samples in the group. In contrast, the Num Indv of Alashan Cashmere goats was significantly different from the sample numbers.

Based on the Fst results of every group, the value of Fst between Hainan Black goats and Leizhou goats was 0.0411077, ultimately serving as the lowest Fst (Table 3). This result aligns with the fact that the Hainan Black goat is related to the Leizhou goat but is distributed in different places [3]. As the Boer goat is the only foreign goat breed selected, the higher Fst is within our expectations. However, a significant difference was found in the genetic differentiation between the two groups when Fst was >0.15. Among all group comparisons, only the Fst between the Leizhou and Boer goats was higher than 0.15 (0.187433), which indicates that there was no significant difference among all breeds, except between the Leizhou goat and the Boer goat. This result indicates that more distinct genetic variations have appeared in these two breeds, and no significant geographical structuring has occurred in the Chinese goat populations [20].

### 3.4. Population−Specific SNP Annotation and Association Signals Unique to Hainan Black Goat Populations

A total of 88,454,696 SNPs were employed to assess the genetic independence of Hainan Black goats and the other six breeds. To better understand the selection of these populations, the population-specific SNPs in the 82 samples were split into two groups. Hainan Black goat samples were used as one group (popC *n* = 16), while samples from the other six breeds were assigned to the other group (*n* = 66). For example, the highest number of population-specific SNPs was observed in the popC population, followed by other populations (Figure 4A). Despite the pronounced divergence of these populations, this was reflected in the high number of population−specific SNPs. Interestingly, 30 common population−specific SNP genotypes differed at only one locus in the popC group and other groups (Figure 4A) (Table A2). This observation is consistent with the fact that environment and genotype can interact to influence the degree of diversification among populations, as different ancestral genotypes show the greatest diversification across the two surface-type environments [20]. Population−specific SNPs were annotated to 417 genes, effectively annotated to 363 genes in the popC group and 80 genes in the other group. Among the genes with defined functions, five functional categories were highly enriched with differentially expressed products: genes involved in gated channel activity, post-synapse, dendrite development, dendrites, and the cell cortex (Figure 4B). GO analysis revealed the involvement of 3 and 17 genes in dendrite and dendrite development in the other group and popC group. This difference is not surprising as dendrites are components that receive and process neuronal inputs; one of the most important features of neuronal function is the ability to adapt dynamically to changing environments and neuronal activity.

The number of population−specific SNPs per 1 M region of the chromosome was counted. A total of 23 regions with different numbers of population−specific SNPs between the popC group and the other group were detected (*p* < 0.005). The results revealed a higher number of population-specific SNPs in the 21 regions of the chromosome in this group compared to the other groups (Table A3). To identify selective signals in Hainan Black goats, we compared the popC group to the other group by calculating the ratio of nucleotide diversity and pairwise genetic differentiation (Fst) (Figure 4C). Outlier windows supported by two methods (θπ ratio: top 1%, Fst: top 1%) were considered to be breed-specific regions under positive selection. After merging consecutive outlier windows, 21 selected regions showed differences in the number of population-specific SNPs (Figure 4D). Regions containing seven genes were identified, including *CNGB3*, *CNBD1*, *EXOC3L4*, *LOC106501899 (TNFAIP2)*, *LOC108638483*, *TNFAIP2,* and *MARK3* (microtubule affinity regulating kinase 3). *CNBD1* contained a total of 21 SNPs in the popC group and only one SNP in the other group. *TNFAIP2* contained seven SNPs in the popC group. The *TNFAIP2* gene may be the target gene of retinoic acid in acute promyelocytic leukemia, and melanocytes are important target cells of retinoic acid [21]. By analyzing the functional protein association network using a string database, we found that the TNFAIP2 protein interacted with EXOC3L4 and CNGB3 in the popC group in this network (Figure 5A) and the CNBD1 protein interacted with CNGB3 in the other groups (Figure 5B). The above results indicate the identification of selective signals implicated in melanin.

Four breeds were selected from the other groups, which represented the eastern region (Jining Grey goat, popE), the western region (Dazu Black goat, popD), the northern region (Alashan Cashmere goat, popG), and the foreign region (Boer goat, popF). Further, the selection signals in the Hainan Black goat (popC) group were analyzed (Figure 6). These results strongly suggest that 259 highly conserved selective signal regions (Figure 7A) involved 20 genes in the popC group, with *CNBD1* among these 20 genes. Among unique-selected regions, 1265 genes were involved in popC (Figure 7B). Functional enrichment analysis was performed for select regions corresponding to annotated genes in the popC group compared to the popD, E, F, and G groups; the results are shown in Figure 8. The regulation of the melanin biosynthetic process, circadian entrainment, regulation of cAMP-mediated signaling, and the Rap-1 signaling pathway between popC and popG were significantly enriched (*p* < 0.05) (Figure 8D). Interesting observations included the presence of *CDH15*, *ASIP,* and *PARD3* among the melanin biosynthetic process, with strong signatures of selection.

## 4. Discussion

China is located on the western bank of the Pacific Ocean in eastern Asia. The territory is vast, and the terrain is high in the west and low in the east. The terrain has a ladder shape and is rich in goat resources. In the present study, six indigenous goat breeds from four typical regions of China—the Inner Mongolia, south-west agricultural, north China, and south-central agricultural regions, as well as one foreign goat breed—were selected for population genetic analysis (Figure 1). Establishing the genetic relationship between Hainan Black goats and the other six goat breeds contributed to the understanding of the evolutionary history of the Hainan Black goat. The population was analyzed using PCA and NJ-tree. Based on the results, the partitioning of the genetic diversity of the breeds was consistent with their geographic distributions. Population genetic analysis can distinguish between physiological and geographical origins. According to pairwise Fst, the relationships between Hainan Black goats and Leizhou goats were closer than those of other breeds (Table 3); this result was also confirmed using PCA and NJ-tree (Figure 3). An introduction to Chinese varietal records reported that Leizhou goats were distributed in the Leizhou Peninsula and Hainan region [3]. Leizhou goats distributed in the Hainan region are always named Hainan Black goats by local people, which are separated from the Leizhou peninsula by sea. Therefore, Hainan Black goats could be gradually differentiated from Leizhou goats; however, the difference was not statistically significant. Gene flow analysis revealed the gene flow from Leizhou goats to Hainan Black goats, thereby verifying this result. Boer goats, which served as a foreign control group, had greater divergence than Leizhou, Longlin, and Alashan Cashmere goats. Collectively, the results were consistent with those of PCA and NJ-tree.

In this study, several strongly selected candidate genes were identified in Hainan Black goats and six other goat breeds. Further, 21 SNPs were found in *CNBD1* in the Hainan Black goat group and only 1 SNP was found in the other group (Figure 4D). *CNBD1* has previously been implicated in blocking access to the conserved nuclear pore localization signal in *EPAC2A* (catalytic domains of cAMP sensors), reducing its ability to interact with nuclear binding sites and affecting the intracellular location and response to elevations in intracellular cAMP [22]. The dysfunction of *CNBD1* might result in abnormal cAMP signaling, subsequently influencing the occurrence of the disease, which might be the reason for the strong disease resistance of Hainan Black goats in adapting to the hot and humid climate of Hainan. The regulation of cAMP-mediated signaling was also significantly enriched between popC (Hainan Black goat) and popG (Alashan Cashmere goat) (Figure 8D), which further confirmed this speculation. In the current study, *CNGB3* was found to be regulated by *CNBD1*. The *CNGB3* gene encodes the β subunit of the cyclic nucleotide-gated channel in cone photoreceptors; mutations in this gene cause achromatopsia in humans [23,24], which is also observed in animals [25]. However, *CNGB3* is regulated by *CNBD1* and is commonly involved in light sensor regulation. Through light sensor regulation, they may be crucial genes that regulate Hainan Black goat production, from seasonal to nonseasonal. Seventeen genes were found to be involved in dendrite development in the Hainan Black goat group, while only three genes were found in the other group (Figure 4B). Goats show a seasonal pattern in reproductive activity related to annual variations in the photoperiod [26]. Further, changes in day length are less pronounced in equatorial, tropical, and subtropical regions. Seasonality in reproduction is, therefore, less marked, and most local goats in the tropics can breed all year-round [26]. Therefore, the Hainan Black goat requires more regulation in terms of dendrite development to adapt to the climate of tropical and subtropical regions for nonseasonal production. This notion explains the enrichment of the light sensor genes in the Hainan Black goat population; the longer day length accelerates light sensor gene evolution and enrichment.

In the current study, four and seven SNPs in *LOC106501899* (annotated to *TNFAIP2*, respectively) were observed in the Hainan Black goat group (Figure 4D). *TNFAIP2* plays an essential role in inflammation, angiogenesis, cell proliferation, adhesion, migration, apoptosis, and proliferation. The expression of *TNFAIP2* is regulated by multiple transcription factors and signaling pathways, including NF-κB, KLF5, and retinoic acid [27,28]. In summary, *TNFAIP2* is a primary response gene induced by multiple proinflammatory molecules at the transcriptional activation level. The expression of *TNFAIP2* is frequently abnormal in human cancers and infectious diseases [27]. SNPs in the 3′-UTR of *TNFAIP2* have been linked to several diseases. Therefore, Hainan Black goats might have stronger disease resistance to better adapt to the natural hot and humid climate of the Hainan region. The copy-number variation regions (CNVs) in *EXOC3L4* (exocyst complex component 3 like 4) and *TNFAIP2* (TNF α induced protein 2) genes, which are associated with natural antibodies, are considered candidates for innate host defense and disease resistance in dairy cows [29]. However, *EXOC3L4* had five specific SNPs in the Hainan Black goat group. This result further confirmed the hypothesis of stronger disease resistance in Hainan Black goats.

*CDH15*, *ASIP*, and *PARD3* were strong signatures of selection between popC (Hainan Black goat) and popG (Alashan Cashmere goat) (Figure 7B) and are among the regulation of melanin biosynthetic processes with strong selective signatures (Figure 8D). Four different CNVs related to the coat color of goats were located at the *ASIP* locus. They were associated with four different types of coat color patterning in seven Swiss goat breeds [30]. In another study, *ASIP* was also confirmed to control several breed-specific coat color patterns [31], which explains *ASIP* as the strong signature of selection between Hainan goats and white Alashan goats with different colors.

Traditionally, photoprotection is believed to be the purpose of melanogenesis or skin pigmentation [32]. In the Hainan region, Hainan Black goats require the biosynthesis of more melanin, which functions as a broadband UV absorbent, an antioxidant, and a radical scavenger. Additionally, the Hainan Black and Alashan Cashmere goats have black and white coat colors, respectively, which reasonably explains the enrichment in the melanin biosynthetic process and the lack of enrichment between popC (Hainan Black goat) and popD (Dazu Black goat) groups and popE (Jining Grey goat) and popF (Boer goat) groups. Based on the appearance of the different goats, the popD, pope, and popF groups presented different colors. However, Hainan Black goats require more pigmentation to increase the resistance of their skin to intense sunshine in the Hainan region, which has a tropical and subtropical environment that introduces UV radiation and sunburn. Melanocytes are phagocytic cells that play a role in inflammatory responses. Melanin has also been found to counteract the damage caused by reactive oxygen species in apoptotic tissues [33]. Melanocytes produce other substances that regulate and enable crosstalk between the different cell types in the epidermis. α-MSH and ACTH peptides produced in the epidermis induce NO production in melanocytes. Moreover, melanocortins may regulate the release of cytokines, catecholamines (CA), and serotonin (5HT) from melanocytes [34]. In summary, melanogenesis in Hainan Black goats contributes to their adaptation to local climate and geography, especially their resistance to skin diseases.

Circadian entrainment was enriched between Hainan Black and Alashan Cashmere goats. Evidently, the Alashan Cashmere goat has traditional seasonality in production, while Hainan Black goats tend to breed year-round. The difference in the reproductive performance of the two goat breeds was mainly caused by circadian entrainment changes introduced by day-length changes in the two regions [26]. Differential gene expression analysis of the 10 clusters of cumulus cells obtained from 10 cumulus–oocyte complexes from 10 patients was performed. Circadian entrainment is the most affected pathway in clinical pregnancy [35], which indirectly indicates that circadian entrainment affects the reproductive cycle. According to previous studies, the Rap-1 signaling pathway has an impact on reproduction [36,37]. In the current study, the Rap-1 signaling pathway was found to be enriched in Hainan Black and Alashan Cashmere goats. The pathway may regulate the maturation of the oocyte, thereby affecting the reproduction of goats that are adaptive to the different environments in the Hainan and Inner Mongolia regions. In a recent study, progesterone-mediated oocyte maturation was reported as a significant pathway in goose ovarian tissue across different reproductive periods [38]. Progesterone-mediated oocyte maturation was enriched between the Hainan Black goat and the Alashan Cashmere goat, indicating that the Hainan Black goat and the Alashan Cashmere goat have evolved different reproductive traits to adapt to their local environment.

## 5. Conclusions

In summary, our comparative genomic analyses provide new insights into the diversity and selective signals in Hainan Black goats and the genetic relationships between Hainan Black goats and the other six goat breeds. Selective sweep analysis revealed some interesting candidate genes and pathways affected by natural or artificial selection involving reproductive or productive traits, immune resistance to disease, the melanin biosynthetic process, and light sensitivity. Elucidating their genomic diversity will provide basic material for the conservation and utilization of germplasm resources in Hainan Black goats. Our results provide important insights into the genomic selection signature of local goat breeds in the Hainan region. The establishment of a genetic relationship between Hainan Black goats and the other six goat breeds will help us understand the evolutionary history of Hainan Black goats. Identifying selective signals will not only help further investigate the genetic mechanism underlying the specific characteristics but also provide candidate SNPs and molecular gene targets for the corresponding important traits of Hainan Black goats, which are helpful for further exploration of the germplasm resources of the Hainan Black goat breed.

## Figures and Tables

**Figure 1 genes-13-01539-f001:**
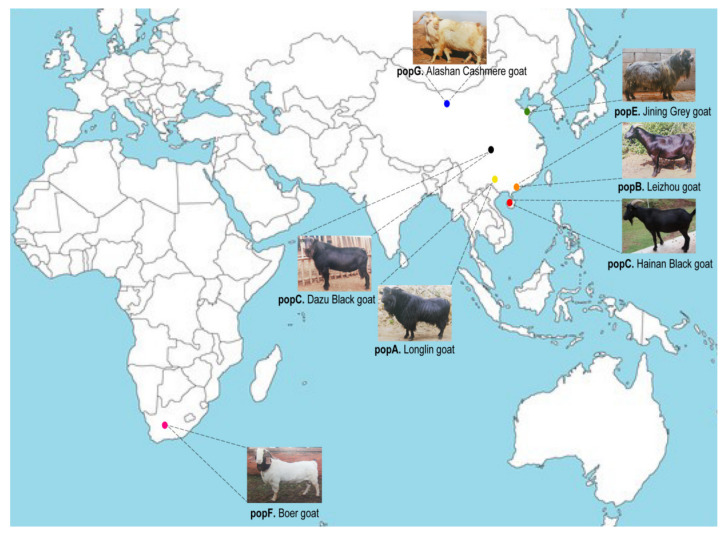
Summary of all seven goat breeds included in this study. Geographic map indicating the distribution of the goats included in this study. Goat photographs reprinted from Ref. [3] except Hainan Black goat photograph was taken by Wencan Zhang.

**Figure 2 genes-13-01539-f002:**
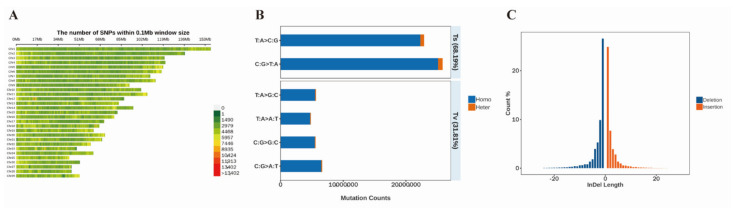
SNP detection and distribution. (**A**) Distribution map of SNPs on chromosomes; the number of SNPs within the 0.1 Mb window size. The color gradient from green to red indicates the number of SNPs. (**B**) SNP mutation spectrum. Blue indicates that the SNPs were homozygous, and orange indicates that the SNPs were heterozygous. (**C**) InDel length distribution. The length of each bar represents the number of indels. Blue represents deletions, and orange represents insertions.

**Figure 3 genes-13-01539-f003:**
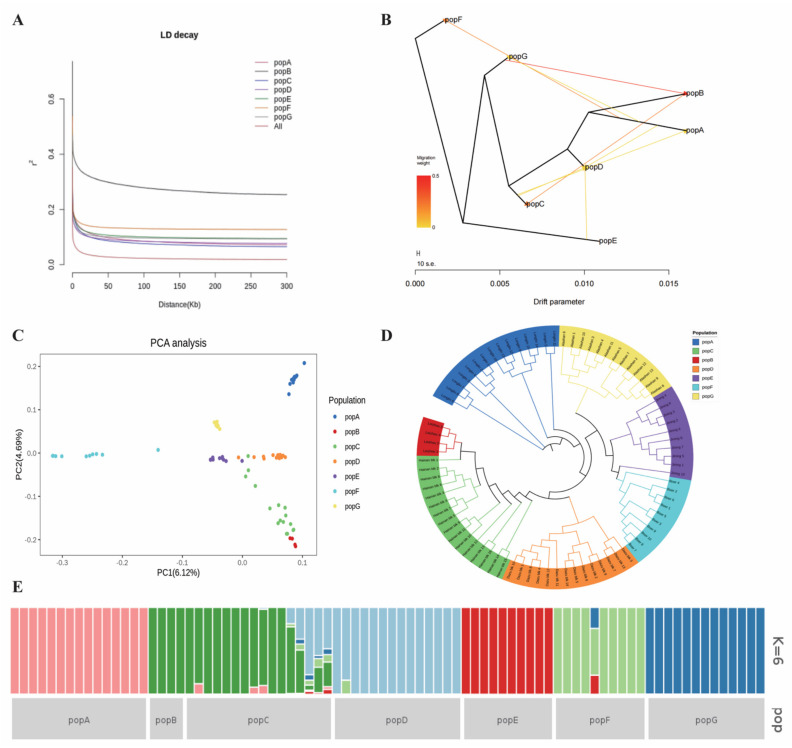
Genetic structure and characterization of 82 goats from 7 breeds. (**A**) LD analysis of each goat breed as well as all goats. (**B**) Pattern of population splits and mixture between the seven populations. The scale bar depicts 10 times the average standard error of the estimated entries in the sample covariance matrix. The migration edge from one lineage into another is colored according to the percent ancestry received from the donor population. (**C**) Principal components analysis (PCA) of seven goat breeds using their first two components. (**D**) Phylogenetic tree analysis of 82 individuals based on autosomal SNPs. (**E**) Ancestry proportions of 82 individuals using K = 6 clusters. popA (Longlin goat), popB (Leizhou goat), popC (Hainan Black goat), popD (Dazu Black goat), popE (Jining Gray goat), popF (Boer goat), and popG (Alashan Cashmere goat).

**Figure 4 genes-13-01539-f004:**
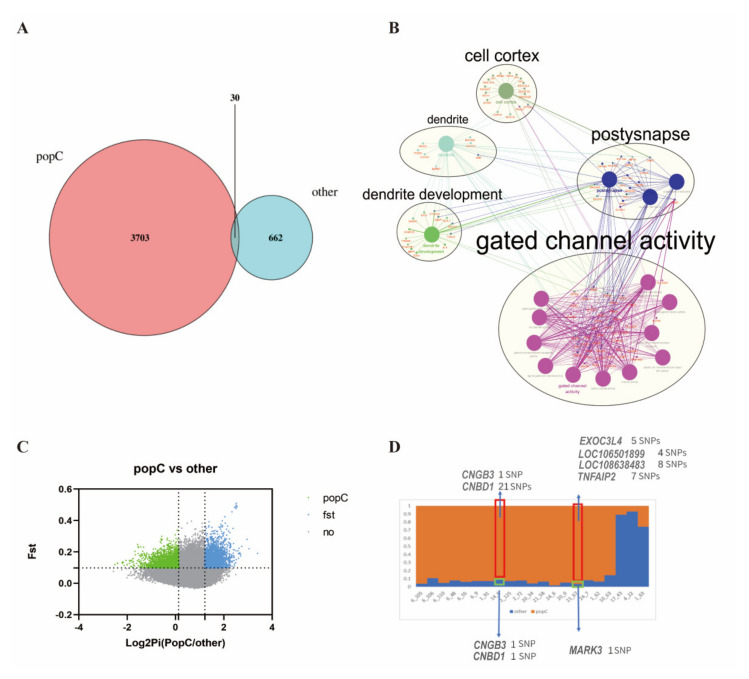
Population−specific SNP annotation and selective signals in Hainan Black goats. (**A**) Venn diagram of the population−specific SNPs in the popC group (Hainan Black goat) and the other group. (**B**) Functional enrichment analysis of 417 genes annotated by the population−specific SNPs. The bigger dot in the circle represents the function cluster, and the smaller dot and triangle in the circle represent genes in popC and other group, respectively. (**C**) Selective clearance analysis between the popC group (Hainan Black goat) and the other group (θπ ratio: top 1%, Fst: top 1%). (**D**) Proportion of selected regions in the number of population−specific SNPs. Orange represents popC, and blue represents the other group. The abscissa represents the selected regions, chromosome regions with the sliding windows of 1 Mb.

**Figure 5 genes-13-01539-f005:**
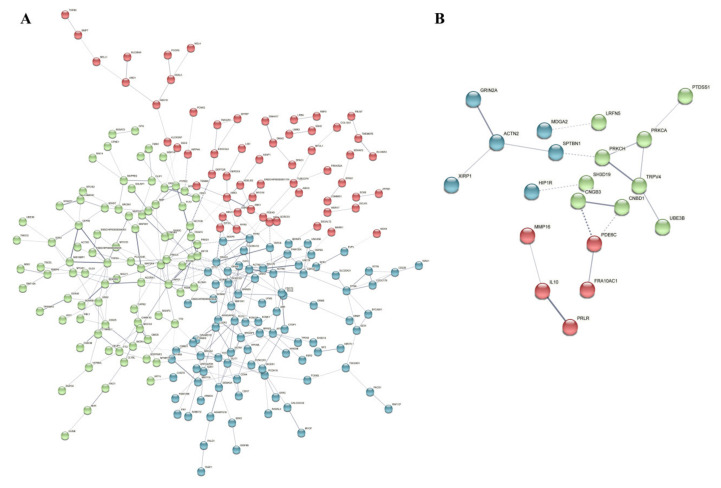
Protein–protein interaction (PPI) network analysis. (**A**) PPI network analysis of population specific SNP annotated 363 genes in the popC (Hainan Black goat) group. Among them, 235 genes were visualized, and the other genes that were not enriched into any network were not visualized. (**B**) PPI network analysis of population specific SNP annotated 80 genes in the other group. Twenty of the genes were visualized, and the other genes that were not enriched into any network were not visualized.

**Figure 6 genes-13-01539-f006:**
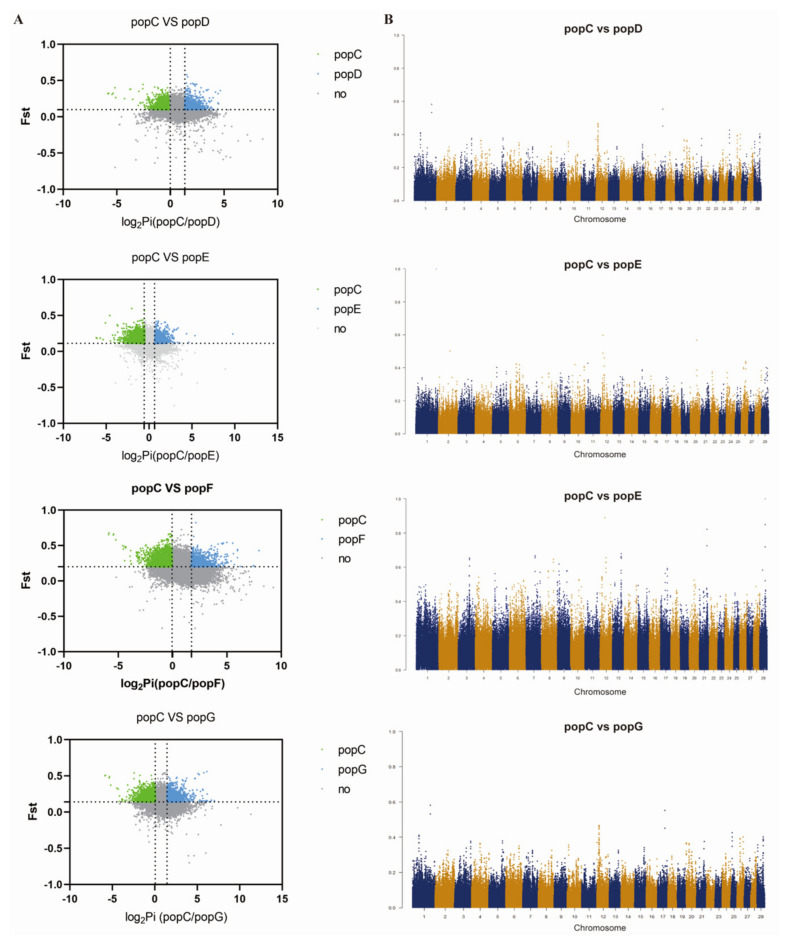
(**A**) Genomic regions with selective sweep signals in the popC (Hainan Black goat) group and the popG (Alashan Cashmere goat), popF (Boer goat), popE (Jining Grey goat), and popD (Dazu Black goat) groups. Distribution of the θπ ratios (θπ, domestic/θπ, Tibetan) and Fst values, which are calculated in 20 kb window, sliding in 5 kb steps. Data points located to the left and right of the left and right vertical dashed lines, respectively (corresponding to the 5% left and right tails of the empirical θπ ratio distribution) and points above the horizontal dashed line (the 5% right tail of the empirical Fst distribution) were identified as the selected regions. The data point in green represents genomic regions with strong selective sweep signals in the popC (Hainan Black goat), the data point in blue represents genomic regions with strong selective sweep signals in each of the other groups compared with the popC group, and the data point in grey represents genomic regions without strong selective sweep signals. (**B**) Overview of the selective sweeps in the popC (Hainan Black goat) group and the popG (Alashan Cashmere goat), popF (Boer goat), popE (Jining Grey goat), and popD (Dazu Black goat) groups. Yellow and dark blue areas are used to distinguish selective sweep signals between different chromosomes.

**Figure 7 genes-13-01539-f007:**
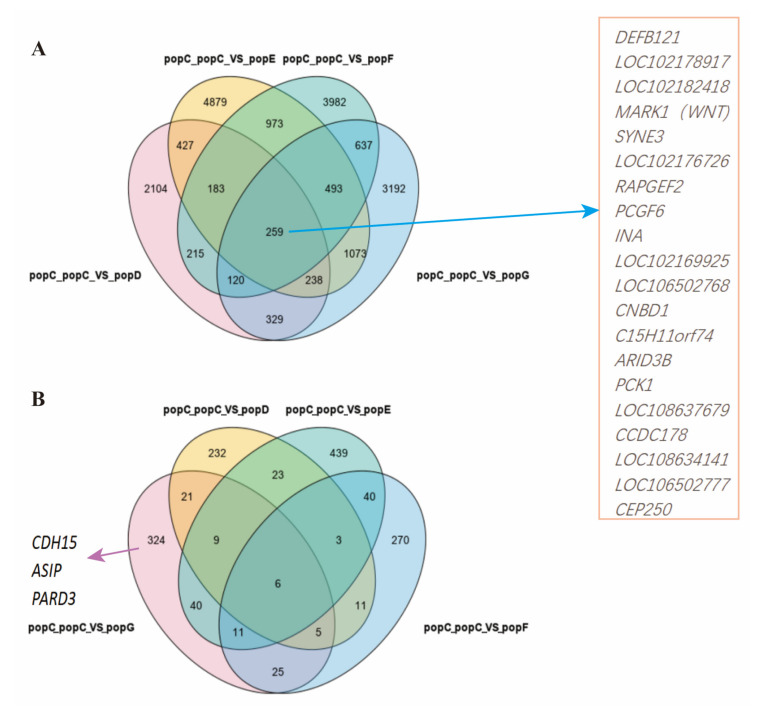
(**A**) Venn diagram of the Hainan Black goat selective signals regions. A total of 20 genes are listed in the red box and are genes in the selection signal region of the Hainan Black goat between popC (Hainan Black goat) and popD (Dazu Black goat), popE (Jining Grey goat), popF (Boer goat), and popG (Alashan Cashmere goat). (**B**) Venn diagram of the Hainan Black goat selective genes. *CDH15*, *ASIP,* and *PARD3* are melanin-related genes and are Hainan Black goat selective genes between the popC (Hainan Black goat) group and the popG (Alashan Cashmere goat) group. PopC_popC vs. popE, PopC_popC vs. popE, PopC_popC vs. popF, and PopC_popC vs. popE represent (compared to popD, PopE, and_popF) the selective signal regions or selective genes in popC.

**Figure 8 genes-13-01539-f008:**
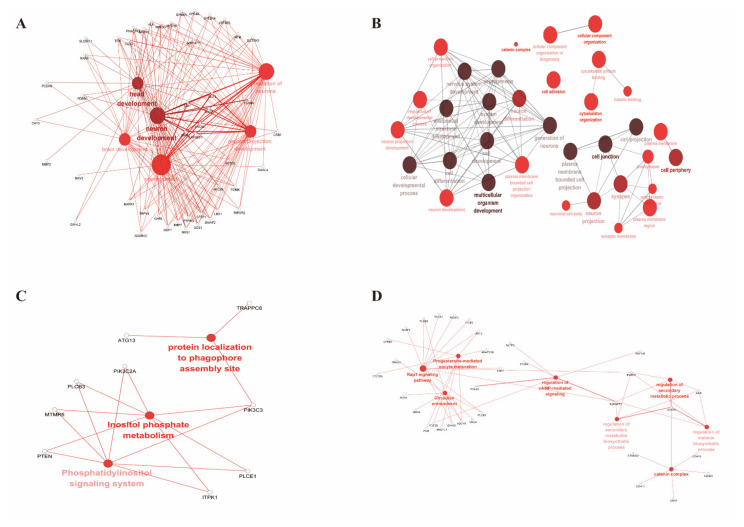
Functional enrichment analysis of the genes that were correspondingly annotated based on the selected regions significantly enriched between popC (Hainan Black goat) and popD (Dazu Black goat), popE (Jining Grey goat), popF (Boer goat), and popG (Alashan Cashmere goat), respectively (*p* < 0.05). Solid dots and hollow dots represent the function and gene, respectively; the color and size of solid dots represents the degree of genes in the correlation network, that is, the larger and the deeper color of the solid dots represent the more genes associated with the function.

**Table 1 genes-13-01539-t001:** Sample data information.

Group	Breed	Number	Origin	BioProject
popA	Longlin goat	15	South-central agricultural regions of China	PRJNA631433
popB	Leizhou goat	5	South-central agricultural regions of China	PRJNA399234
popC	Hainan Black goat	16	South-central agricultural regions of China	PRJNA754269 *
popD	Dazu Black goat	16	South-west agricultural of China	PRJNA479946
popE	Jining Grey goat	10	North China	PRJNA560446
popF	Boer goat	10	South Africa	PRJEB25062
popG	Alashan Cashmere goat	15	Inner Mongolia of China	PRJNA338022

* Result based on unpublished sequencing data.

**Table 2 genes-13-01539-t002:** Statistical results of the genetic diversity of each population.

Population	Num Sam	Num Indv	Obs Het	Exp Het	Pi	Fis
Longlin goat	15	11.3478	0.1808	0.2418	0.2531	0.196
Leizhou goat	4	3.3589	0.1772	0.1858	0.2204	0.0822
Hainan Black goat	16	14.9233	0.2278	0.2667	0.276	0.1448
Dazu Black goat	14	10.8296	0.1851	0.2596	0.2727	0.237
Jining Grey goat	10	8.2422	0.2119	0.2637	0.2811	0.1778
Boer goat	10	5.5103	0.1778	0.2073	0.2307	0.1201
Alashan Cashmere goat	13	6.5266	0.1872	0.2419	0.2638	0.1805

**Table 3 genes-13-01539-t003:** Statistics of the Fst results between populations.

	Leizhou Goat	Hainan Black Goat	Dazu Black Goat	Jining Grey Goat	Boer Goat	Alashan Cashmere Goat
Longlin goat	0.116601	0.075297	0.0730351	0.0824533	0.126503	0.0868544
Leizhou goat		0.0411077	0.0879101	0.0992719	0.187433 *	0.13665
Hainan Black goat			0.0514138	0.0552972	0.0908222	0.0682161
Dazu Black goat				0.0650801	0.106479	0.0759338
Jining Grey goat					0.0945023	0.070164
Boer goat						0.115508

* The Fst between Leizhou and Boer goats was higher than 0.15.

## Data Availability

The sequence data of the Hainan Black goat was deposited at GenBank under accession number PRJNA754269.

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
