# Peer review of "Whole-Genome Sequencing Reveals the Genomic Characteristics and Selection Signatures of Hainan Black Goat"

_genes, 2022, doi:10.3390/genes13091539_

Round 1
Reviewer 1 Report
This study aimed to reveal the selection signatures associated with important traits in seven goat breeds.
The English of the manuscript needs major revision by a native person.
In addition, it seems that some breeds of the studies by Guo et al. (2018) [Whole-genome sequencing reveals selection signatures associated with important traits in six goat breeds] and Chen et al. (2021) [Whole-genome resequencing reveals diversity and selective signals in Longlin goat] were similar to the breeds considered in this study.
The authors need to clarify the difference between their own study with the abovementioned studies.
Author Response
Point 1: This study aimed to reveal the selection signatures associated with important traits in seven goat breeds.
Response 1: Yes, the aim of the study was to revealed the selection signatures associated with important traits in seven goat breeds, especially the Hainan Black goat.
Point 2: The English of the manuscript needs major revision by a native person.
Response 2: The language of manuscript has been revised extensively as seen in the revised version.
Point 3: In addition, it seems that some breeds of the studies by Guo et al. (2018) [Whole-genome sequencing reveals selection signatures associated with important traits in six goat breeds] and Chen et al. (2021) [Whole-genome resequencing reveals diversity and selective signals in Longlin goat] were similar to the breeds considered in this study.The authors need to clarify the difference between their own study with the above mentioned studies.
Response 3: On the one hand, Hainan black goat, as one of the most important local breed in Hainan, as the research object, which was the first time selected and researched in our study as one of representative tropical and subtropical areas goat breed and explore its important characteristics through comparison with other six goat breeds. On the other hand, the different geographical location and climate of China are obviously different. We selected goat breeds from the four typical regions of China for comparison, including Alashan Cashmere goat in Inner Mongolia region, Dazu Black goat in southwest agricultural region, Jining Grey goat in North China region and Longlin goat and Leizhou goat in south-central agricultural region, and a foreign breed (Boer goat) from South Africa was also included as a control breed. Our aim was to explore the specific characteristics of different goats to reveal genetic loci under artificial and natural selection in different environments. However, in the study of Guo et al. (2018), they mainly explored and compared five different goat breeds in Southwest China and one foreign goat breed, including Boer, Meigu, Jintang Black, Nanjiang Yellow, Tibetan and Tibetan cashmere goat [1]. Among them, except Boer goats, other goat breeds are different from our study, and the goats they selected are all located in southwest of China. In another study of Chen et al. (2021) [2], the authors analyzed the important performance of Longlin goat in Guangxi by comparing Longlin goat with other goat breeds in the same latitude. Therefore, the purpose of our article are different from those of the other two articles. Additionally, to highlight the purpose of our study, we modified the title of manuscript and Figure 1 as following:
Title: Whole-genome Sequencing Reveals the Genomic Characteristics and Selection Signatures of Hainan Black Goat
Figure 1. Summary of all seven goat breeds included in this study. Geographic map indicating the distribution of the goats included in this study.
References
[1] Guo J, Tao H, Li P, Li L, Zhong T, Wang L, Ma J, Chen X, Song T, Zhang H. Whole-genome sequencing reveals selection signatures associated with important traits in six goat breeds. Sci Rep. 2018 Jul 10;8(1):10405. doi: 10.1038/s41598-018-28719-w.
[2] Chen Q, Huang Y, Wang Z, Teng S, Hanif Q, Lei C, Sun J. Whole-genome resequencing reveals diversity and selective signals in Longlin goat. Gene. 2021 Mar 1;771:145371. doi: 10.1016/j.gene.2020.145371.
Reviewer 2 Report
abstract: lines 14-15; number of goats and cows in the world are similar (one billion each), but it is worth of mentioning, that goats population doubled during last four decades and 90% of goats are found in developing countries, and Asia has the largest population. This manuscript is worth of mentioning: https://doi.org/10.3389/fvets.2021.648500
Last sentence of the abstract is too general.
line 60: 5 breeds
Please add more information about DNA extraction, and most of all selected animals.
Table 1: number of individuals; and number of individuals used for the experiment is sufficient
some Figures legends are too brief
Keeg analysis is not presented in results, just the figure and 1 sentence.
Do the Authors have the clinical data about better disease resistance in Hanaina goats? Or in general, health data? what is the medum life lenght of this breed? fertility?
conclusions are too-far reaching. crossbreeding to avoid inbread? or crossbreeding on which features?
Author Response
Point 1: abstract: lines 14-15; number of goats and cows in the world are similar (one billion each), but it is worth of mentioning, that goats population doubled during last four decades and 90% of goats are found in developing countries, and Asia has the largest population. This manuscript is worth of mentioning: https://doi.org/10.3389/fvets.2021.648500
Last sentence of the abstract is too general.
The abstract was modified as following according to the suggestions of Reviewer and the manuscript (https://doi.org/10.3389/fvets.2021.648500) was cited in the Introduction part:
Abstract: Goats have become one of the most adaptive and important liverstock species that distributed mainly in developing countries in recent years. The Hainan Black goat is a native goat breed of the Hainan region, which is generally well-liked by the local population and is thus raised in large numbers. However, the genomic diversity and selective signals of Hainan Black goat have not been clearly elucidated yet. Therefore, in this study, we performed whole genome resequencing of 16 Hainan Black goats and compared the results with those of 71 goats of 6 other breeds from different geographic regions. Principal component analysis (PCA) and phylogenetic analysis identified 7 lineages for all goats. Hainan Black goats showed most similarity with Leizhou goats and least similarity with Boer goats. Selective sweep analysis identified candidate genes associated with various functions, including immune resistance to disease (TNFAIP2 and EXOC3L4), melanin biosynthetic process (CDH15, ASIP and PARD3) and light sensitivity (CNGB3 and CNBD1), underlying strong selection signatures in Hainan Black goats. Melanin biosynthetic process, circadian entrainment, regulation of cAMP-mediated signaling, Rap-1 signaling pathway were significantly enriched in Hainan Black and Alashan Cashmere goats. This result may be important for understanding trait of each goat breed. Selection signature analysis revealed candidate SNPs and genes correlated with the trait of Hainan Black goat. Collectively, our results provide valuable insights into the genetic basis of specific traits correlated with the Hainan island climate and artificial selection in certain local goat breeds, and importance of protecting the breed resources.
Intoduction: Asia has the largest world population of goats [1]. In particular, China has abundant goat resources and a large goat population due to the presence of diverse ecosystems. In 2020, the total number of goats being raised in China was 133.45 million [2].
(1)Utaaker KS,; Chaudhary S,; Kifleyohannes T,; et al. Global Goat! Is the Expanding Goat Population an Important Reservoir of Cryptosporidium? Front Vet Sci 2021, 8, 648500. https://doi.org/10.3389/fvets.2021.648500.
(2)National Bureau of Statistics of China. Available online: http://www.stats.gov.cn/tjsj/ndsj/ (accessed on July 1, 2022).
Point 3: line 60: 5 breeds
Corrected to “five goat breeds”.
Point 4: Please add more information about DNA extraction, and most of all selected animals.
The information was added as following:
Genomic DNA was collected from the ear tissues of 16 Hainan Black goats using the standard phenol-chloroform extraction method, and subjected to electrophoresis (in 1.2% agarose gel) and UV spectrophotometry.
Point 5: Table 1: number of individuals; and number of individuals used for the experiment is sufficient. some Figures legends are too brief
The figures legends were added in details as following:
Figure 1. Summary of all seven goat breeds included in this study. Geographic map indicating the distribution of the goats included in this study (Photographs from Animal genetic resources in China: Sheep and Goats [3].
Figure 2. SNP detection and distribution. (A) Distribution map of SNPs on chromosome; the number of SNPs within the 0.1-Mb window size. The color gradient from green to red indicates the number of SNPs. (B) SNP mutation spectrum. Blue indicates that the SNPs were homozygous and orange indicates that the SNPs were heterozygous. (C) Indel length distribution. Distribution of InDels. The length of each bar represents the number of InDels. Blue represents deletions and orange represents insertions.
Figure 3. Genetic Structure and Characterization of 82 goats from 7 breeds. (A) LD analysis of each goat breed as well as all goats. (B) Pattern of population splits and mixture between the seven populations. The scale bar depicts 10 times the average standard error of the estimated entries in the sample covariance matrix. The migration edge from the one lineage into another is colored according to the percent ancestry received from the donor population. (C) Principal components analysis (PCA) of seven goat breeds using their first two components. (D) Phylogenetic tree analysis of 82 individuals based on autosomal SNPs. (E) Ancestry proportions of 82 individuals using K = 6 clusters. popA (Longlin goat), popB (Leizhou goat), popC (Hainan Black goat), popD (Dazu Black goat), popE (Jining Gray goat), popF (Boer goat) and popG (Alashan Cashmere goat).
Figure 4. Population specific SNPs annotation and selective signals in Hainan Black goat. (A) Venn diagram of the population-specific SNPs in popC and the other group. (B) Functional Enrichment analysis of 417 genes annotated by the population-specific SNPs. (C) Selective clearance analysis between popC group (Hainan Black goat) and the other groups (θπ ratio: top 1%, Fst: top1%). (D) Proportion of selected regions in the number of population-specific SNPs. Orange represents popC and blue represents the other group. The abscissa represents the selected regions, chromosome regions with the sliding windows of 1Mb.
Figure 5. Protein-protein Interaction (PPI) networks analysis. (A) PPI network analysis of genes in the popC (Hainan Black goat) group. (B) PPI network analysis of genes in the other group.
Figure 6. (A) Genomic regions with strong selective sweep signals in the popC (Hainan Black goat) group and the popG (Alashan Cashmere goat), popF (Boer goat), popE (Jining Grey goat), popD (Dazu Black goat) groups. Distribution of the θπ ratios (θπ, domestic/θπ, Tibetan) and Fst values, which are calculated in 20-kb window sliding in 5-kb steps. Data points located to the left and right of the left and right vertical dashed lines, respectively (corresponding to the 5% left and right tails of the empirical θπ ratio distribution), and points above the horizontal dashed line (the 5% right tail of the empirical Fst distribution) were identified as the selected regions. (B) Overview of the selective sweeps in the popC (Hainan Black goat) group and popG (Alashan Cashmere goat) , popF (Boer goat), popE (Jining Grey goat), popD (Dazu Black goat) groups.
Figure 7. (A) Venn diagram of the Hainan Black goat selective signals regions. A total of 20 genes are listed in the red box and are genes in the selection signal region of Hainan Black goat between popC (Hainan Black goat) and popD (Dazu Black goat), popE (Jining Grey goat), popF (Boer goat) and popG (Alashan Cashmere goat). (B) enn diagram of the Hainan Black goat selective genes. CDH15, ASIP and PARD3 are melanin related genes and are Hainan Black goat selective genes between popC (Hainan Black goat) group and popG (Alashan Cashmere goat) group. PopC_popC vs popE, PopC_popC vs popE, PopC_popC vs popF, PopC_popC vs popE reperesents that compared to popD, PopE, and_popF, the selective signals regions or selective genes in popC.
Figure 8. Functional enrichment analysis of the genes that was correspondingly annotated based on the selected regions significantly enriched between popC (Hainan Black goat) and popD (Dazu Black goat), popE (Jining Grey goat), popF (Boer goat) and popG (Alashan Cashmere goat), respectively (P < 0.05).
Point 6: Keeg analysis is not presented in results, just the figure and 1 sentence.
Thank you for your suggestion.
The figure 8 was functional enrichment analysis, not only the KEGG pathway analysis. The legend of Figure 8 was modified as following:
Figure 8. Functional enrichment analysis of the genes that was correspondingly annotated based on the selected regions significantly enriched between popC (Hainan Black goat) and popD (Dazu Black goat), popE (Jining Grey goat), popF (Boer goat) and popG (Alashan Cashmere goat), respectively (P < 0.05).
Point 7: Do the Authors have the clinical data about better disease resistance in Hanaina goats? Or in general, health data? what is the medum life lenght of this breed? fertility?
According to the study of Shi et al., the average birth weight, weaning weight at 3 months, weight at 6 months and average body weight at one year old of Hainan black goats were 2.50kg, 8.00kg, 15.41kg and 25.50kg respectively [1]. The average weight of adult male and female Hainan Black goat was 50.00kg and 43.00kg [2].
The sexual maturity of Hainan black goat is early, and the lambs show sexual behavior three months after birth. They are sexually mature at the age of 5-6 months, and can be mated at the age of 8-10 months. The utilization life of breeding sheep is generally 5-8 years. The estrus cycle of ewes is 16-20 days, lasting 48-72 hours, the gestation period is 146-151 days, and the lactation period is generally 2.5 months. Hainan black goat can estrus all the year round, and there are many estrus breeding in August and September each year. Most of the first lambs were single lamb, and the probability of double lambs of the ewes was increased, and the lambing rate was 155%.
(1)Shi, L.; Zhou, X.; Zhou H.; et al. Germplasm Characteristics of Hainan black goat [J]. The Chinese Livestock and Poultry Breeding 2016, 11, 70-71.
(2)Ma, N.. Germplasm Characteristics and ecological characteristics of Hainan black goat [J]. China Herbivore Science 2006, 01, 64+22.
Point 8: conclusions are too-far reaching. crossbreeding to avoid inbread? or crossbreeding on which features?
Conclusion was modified as following:
In summary, our comparative genomic analyses provide new insights into the diversity and selective signals in Hainan Black goats, and the genetic relationships between Hainan Black goats and the other six goat breeds. Selective sweep analysis revealed some interesting candidate genes and pathways affected by natural or artificial selection involving reproductive or productive traits, immune resistance to disease, melanin biosynthetic process and light sensitivity. Elucidating the genomic diversity will provide basic materials for the conservation and utilization of germplasm resources in Hainan Black goats. Our results provide important insights into the genomic selection signature of local goat breeds in the Hainan region. The establishment of a genetic relationship between Hainan Black goats and the other six goat breeds will help us understand the evolutionary history of Hainan Black goats. Identifying selective signals will not only help further investigate the genetic mechanism underlying the specific characteristics, but also provide candidate SNPs and gene molecular targets for the corresponding important traits of Hainan Black goats, which is helpful for further exploration of the germplasm resources of the Hainan Black goat breed.
Round 2
Reviewer 1 Report
The authors revised the manuscript appropriately according to my comments.
Author Response
The manuscript was checked and uploaded.